# Level of Service Model of the Non-Motorized Vehicle Crossing the Signalized Intersection Based on Riders’ Perception Data

**DOI:** 10.3390/ijerph19084534

**Published:** 2022-04-09

**Authors:** Xiaofei Ye, Yi Zhu, Tao Wang, Xingchen Yan, Jun Chen, Bin Ran

**Affiliations:** 1Ningbo Port Trade Cooperation and Development Collaborative Innovation Center, Faculty of Maritime and Transportation, Ningbo University, Ningbo 315211, China; yexiaofei@nbu.edu.cn; 2Faculty of Maritime and Transportation, Ningbo University, Ningbo 315211, China; zhuyi19991125@163.com; 3School of Architecture and Transportation, Guilin University of Electronic Technology, Guilin 541004, China; 4College of Automobile and Traffic Engineering, Nanjing Forestry University, Nanjing 210037, China; xingchenyan.acad@gmail.com; 5School of Transportation, Southeast University, Nanjing 211189, China; chenjun@seu.edu.cn; 6Department of Civil and Environmental Engineering, University of Wisconsin-Madison, Madison, WI 53706, USA; bran@seu.edu.cn

**Keywords:** traffic engineering, non-motorized vehicles, delay, level of service

## Abstract

This article aims to analyze the factors affecting the LOS (level of service) of non-motorized vehicles crossing the signalized intersection and to construct an appropriate method to evaluate the LOS. Aiming at the mixed non-motorized traffic flow of electric vehicles and bicycles in the Chinese metropolis, the delay model in the highway capacity manual (HCM) was modified by taking two new factors into account: the pedestrian traffic rule compliance rate and the fuzzy perception of arrival rate in reality. The results show that the data obtained by the modified model are more consistent with the actual one. Next, a comparison was established between the linear regression method and cumulative logistic regression to determine the variables that affect the LOS, and finally, a LOS evaluation index system based on threshold schemes was defined. The recommended LOS model can provide corresponding references for traffic engineers who seek to improve the level of service in urban intersections.

## 1. Introduction

Due to high convenience and low carbon emissions, non-motorized vehicles have become one of the major sustainable trip modes in developing countries, especially in China. For example, the average trips of non-motorized vehicles occupied 33.2% and 30.4% in the cities of Nanjing and Ningbo, respectively. With the rapid development of shared bicycles and electric bikes (e-bikes), a cumulative 23 million shared bicycles and e-bikes were put into the trip market from 2015 to now. Moreover, non-motorized vehicles are often used as transfer modes along the metro and bus lines to better facilitate the travel of passengers. Obviously, the proportion of bicycle and e-bike trips becomes larger in most Chinese cities. The high proportion of non-motorized trips brings some issues about traffic efficiency and safety. Especially at road intersections, the riders of bicycles and e-bikes cross the street against the traffic signal and are threatened by the high risk of traffic accidents.

Traffic intersections are the most complex locations as different traveling directions are desired by different models of travel, such as motorized vehicles, non-motorized vehicles, and pedestrians [1]. All of them want to prioritize themselves because of the lack of road resources which leads to conflicts between them. In the design and practice of signal timing of intersection, the green-signal time is often shared by the same direct movements of pedestrians, vehicles, and non-motorized vehicles. However, bicycles and e-bikes often invade the parallel crosswalk of pedestrians or the vehicular lane and collide with them [2]. The invasion behavior of non-motorized vehicles will inevitably result in a higher risk of traffic conflicts and lower traffic efficiency. The signal timing should take the crossing behavior of invasion into account. In fact, the minimum green-light time provided at an intersection is affected by the need for sufficient time for bicycles and e-bikes to pass through safely. In turn, the green light signal is assigned to another specific movement, which affects the capacity and delay of the intersection.

In addition, a large number of non-motorized vehicles in China has led to lower intersection service level and longer delays. Level of service (LOS) is defined as a qualitative measure to describe the state of an intersection based on some measurable characteristics (e.g., speed of passing through, actual travel time, traffic delay, and so on). Generally speaking, the delay of an intersection is one of the most important parameters for evaluating the LOS of non-motorized vehicles at signalized intersections. Unfortunately, the current LOS methods were formulated for pure traffic flow conditions and traditional delay and LOS models for non-motorized vehicles without considering the crossing behavior and mixed traffic flow, especially the invading behavior and mixed non-motorized traffic flow combing regular bicycles and e-bikes. The applications of traditional LOS methods could not reflect the real situations in mixed traffic conditions in China. The invasion of non-motorized vehicle crossing behavior into other traffic flows could be described by the conflicts among them.

Accordingly, the paper focuses on evaluating the delays of non-motorized vehicles under the specific circumstance of mixed traffic flow and analyzes the LOS of the intersection by considering the conflict between non-motorized vehicles and other behaviors and the perception of non-motorized riders on traffic safety and the efficiency of intersection traffic. The purpose of this research is to evaluate the LOS of non-motorized vehicles at signalized intersections and to comprehensively analyze the influencing factors of LOS based on the conflict and perception data collected from Ningbo, Nanjing, and Guilin, China. The contributions of this article are to improve the delay model in HCM to make it suitable for the current situation of mixed electric vehicles and bicycles at intersections in China and to apply the method of combining cyclist perception and experimental data for modeling LOS of the non-motorized vehicle crossing the signalized intersection. The findings could further improve and better design the crossing facilities at the signalized intersections.

## 2. Literature Review

LOS as an effective method is applied to design and manage the signal timing and crossing facilities of intersections. It contains several measurable characteristics and the delay is one of the most significant factors which deserve further study.

Related studies on delay and LOS methods focused on the pure bicycle traffic flow on the road segment. By improving the bicycle compatibility index model, Zhao (2014) [3] determined the LOS’s classification standard by analyzing the characteristics of bicycle traffic flow and the influencing factors of cycling. Han (2014) [4] established a LOS evaluation method for non-motorized vehicles based on subjective feelings and the number of over-taking maneuvers. Cao (2015) [5] combined the subjective factors with the physiological characteristics of non-motorized vehicle riders based on the subjective investigation of the non-motorized vehicle LOS. The service level evaluation indicators of non-motor vehicles were selected from both subjective and objective aspects; the grading standards for LOS were determined afterward. Based on the subjective perceptions of the bicycle and pedestrian sharing road traffic survey, an LOS evaluation model was established by Yu et al. (2012) [6] and they also proposed a corresponding grading standard. Yan, Wang, Ye et al. (2018) [7] analyzed the major types of abreast riding and overtaking, and used a binomial logistic model to explore the volume threshold for two-abreast riding as well as the suitable clearances required for comfortable overtaking. Jin, Qu, Zhou et al. (2015) [8] proposed a novel bicycle equivalent unit for e-bike estimation through eight traffic flow fundamental diagrams developed for one-way cycleway capacity estimation and implied that the estimated capacity was independent of a cyclist’s gender and age, but increased with the proportion of e-bikes. Yan, Wang, Chen et al. (2019) [9] analyzed the characteristics of the bicycle–passenger conflicts at bus stops and developed a model to predict the number of conflicts accurately.

Various factors, for instance, peak hour factor, motor vehicle flow, large motor vehicle ratio, pavement condition, the effective width of non-motorized vehicle lanes, and the number of collisions per unit road segment, were considered to establish a bicycle LOS model by Elias (2011) [10]. According to Landis, Vattikuti, Ottenberg et al. (2003) [11], benefitting from the innovative “Ride for Science” field data collection event and video simulations of the Florida Department of Transportation, the new Bicycle LOS model for the main streets was established. The data consisted of participants’ perceptions of how well roadways met their needs as they rode selected arterial roadways and/or viewed simulations of those and other roadways. Kang and Lee (2012) [12] developed a bicycle LOS model by considering the user’s level of satisfaction and multiple factors that affect the bicycle LOS. Griswold, Yu, Filingeri et al. (2018) [13] conducted a survey on cyclist habits, preferences, and user experiences to obtain behavior data. Then, they applied behavioral analysis tools as the proof of concept for a new bicycle LOS measure and combined statistics and behavioral analysis to improve the quality of bicycle level of service measures to make decisions driven by empirically measured cyclist preferences. Liu, Homma, and Iki (2019) [14] introduced and compared two methods for evaluating the quality of bicycle facilities, bicycle compatibility index (BCI) and bicycle level of service (BLOS). Beura, Kumar, and Bhuyan (2017) [15] modeled the quality of services offered to bicycles through movement at urban signalized intersections carrying heterogeneous traffic. A total of six attributes of intersection approaches having significant influences on bicycle service quality were identified with the help of Pearson’s correlation analysis and stepwise regression analysis. Majumdar and Mitra (2018) [16] developed a bicyclist perceived level of service (LOS) criteria by expressing ordered probit (OP) models to assess the quality of bicycle travel from the users’ perspective and develop adequate moderation measures. Beura, Chellapilla, and Bhuyan (2017) [17] applied random forest to identify eight important road attributes affecting the bicycle LOS and proposed a suitable model for urban road segments in mid-sized cities under complex situations. Bicycle LOS measures, thresholds, and estimation procedures for off-street paths, designated bicycle lanes on urban streets, and urban street intersections were provided by The 2016 Highway Capacity Manual. As for urban streets intersections, perceived separation from motorized vehicle traffic, motorized vehicle volumes, cross-street width, and presence and utilization of on-street parking were combined by the bicycle LOS model and, according to the regression model, there were six LOS configurations (A through F) included.

The Highway Capacity Manual 2016 defined bicycle delay with Webster’s model which was based on the assumption that there is no bicycle incremental delay or initial queue delay. The formulation is given below:(1)$$d_{b}=\frac{0.5\times C\times {\left(1-\frac{g_{b}}{C}\right)}^{2}}{1-\min\left[\frac{V_{bic}}{C_{b}},1.0\right]\times \frac{g_{b}}{C}}$$
where

$d_{b}$ is the average delay of the bicycle (s),C is cycle length (s),$g_{b}$ is effective green time for the bicycle lane (s)$V_{bic}$ is the bicycle flow rate (bicycle/h), and$C_{b}$ is the capacity of the bicycle lane (bicycle/h).

The previous research focused on the LOS of regular bicycles on the segments. Most bicycle delay models did not take the e-bike into account. Especially, in the 2016 HCM model, the pure bicycle LOS was used for the planning, design, and management of the bicycle facilities.

## 3. Methodology for the Delay Model of the Non-Motorized Vehicle at Signalized Intersections

### 3.1. Data Collection

Since the research objects are the intersections of a typical metropolis in China and the non-motor vehicles passing through, it is necessary to ensure the authenticity and regularity of these two while collecting data. This requires that the selected intersections have typical non-motorized vehicle traffic flow to ensure the authenticity of the data, available crossing facilities to ensure the feasibility of crossing behavior, and of course, signal lights that work normally are also needed. After all, intersections without signal lights only account for a small part of the city, let alone be representative.

Therefore, the data collection sites of this study were selected in 20 entry roads at 10 intersections in different cities in the Chinese mainland: Ningbo, Guilin, and Nanjing. The crossing width and number of lanes of these intersections are different (ranging from 25 m–40 m wide for the crossing width while the number of lanes varies from four to nine), but the common point of these intersections is that they all set special crossing signal phases for non-motorized vehicles to pass through the intersection. Due to the requirements for the amount of analysis data, we choose to collect data in the morning and evening peak times (7:00–9:00; 16:30–18:30) when the traffic flow is large and people have a strong perception about the LOS of crossing the street. The width of the non-motorized lane at the selected intersection ranges from 0–4 m, which is less than a quarter of the road resources occupied by motor vehicles. However, through the subsequent extraction of the data in the video, it is found that the number of right-turning non-motorized vehicles is as high as 2160/h, which is more than four times higher than that of right-turning motor vehicles (the maximum number of right-turning motor vehicles is 450/h). This shows the importance of evaluating the LOS of intersection from the perspective of non-motorized vehicles.

### 3.2. The Non-Motorized Vehicle Delay Model at Signalized Intersection Crosswalks

Compared with the traditional method of dividing the signal timing into red, yellow, and green periods, this experiment only divides the signal into green phase and non-green phase and records the number of non-motorized vehicles passing through the intersection every 60 s. Considering the non-motorized riders’ compliance with traffic rules and the fuzzy perception of vehicles entering the intersection situation, the delay model of The 2016 Highway Capacity Manual is modified, and two new influencing factors, rule compliance degree, *K_C_*, and irregular arrival rate, *K_NU_*, are introduced into the model. The degree of rule compliance is expressed by the proportion of the number of non-motorized vehicles in the stop line in the non-green phase. This should be 100% as we expected that all riders follow the rules. However, there are always some riders who choose to stop ahead of the stopping line, especially at intersections with large traffic flow during commutes. This kind of action not only affects the right turn of motorized vehicles but also “manually” increases the number of non-motorized vehicles stranded during the non-green phase, eventually reducing the LOS of the intersection. Thus, *Kc* was used to take the impact of those “rule breakers” into consideration. The irregular arrival rate is a variable defined by Li (2005) [18], which is determined by the number of non-motor vehicles in the green phase and the total number of non-motorized vehicles.

The updated model is as follows:(2)$$d_{}=\frac{0.5\times C\times K_{C}\times K_{NU}\times {\left(1-\frac{g_{b}}{C}\right)}^{2}}{1-\min\left[\frac{V}{C_{b}},1.0\right]\times \frac{g_{b}}{C}}$$

### 3.3. The Delay Model Results of the Non-Motorized Vehicle

Table 1 shows the data obtained from the field survey. These data are extracted from the video taken by the camera, and the camera erection position is shown in Figure 1. The camera was erected at selected intersections for a continuous week; the camera memory card was replaced twice in the middle, and finally, the data of the captured video were extracted manually. The data in Table 1 are the average value of the data extracted from Monday to Friday with rounded processing of the number of non-motorized vehicles.

The data include intersection signal period, green light phase duration, number of non-motorized vehicles crossing the street, and the number of conflicts related to non-motorized vehicles crossing the street. Conflicts are used to describe the intrusion behavior of non-motorized vehicles crossing the street, which has nothing to do with the model calculation, so they are not included. The actual intersection delay time calculated from these data is different from the value calculated by the HCM initial delay model and Formula (2), which can be clearly seen from either Table 2 or Figure 2, Figure 3 and Figure 4. It is noteworthy that although the results of Formula (2) do not fully agree with the real data, compared with the HCM model with an error rate of 43.34%, the error rate of 7.52% seems to be acceptable, which also proves that the HCM method is not applicable to the mixed non-motor vehicle flow of electric vehicles and non-motorized vehicles. Therefore, we continue to modify the delay model as follows:(3)$$d_{eb}=\frac{0.5\times C\times K_{C}\times K_{NU}\times {\left(1-\frac{g_{b}}{C}\right)}^{2}}{1-\min\left[\frac{1.056\times V_{eb}-186.687}{C_{b}},1.0\right]\times \frac{g_{b}}{C}}$$
(4)$$d_{b}=\frac{0.5\times C\times K_{C}\times K_{NU}\times {\left(1-\frac{g_{b}}{C}\right)}^{2}}{1-\min\left[\frac{3.875\times V_{b}+366.456}{C_{b}},1.0\right]\times \frac{g_{b}}{C}}$$
where C is cycle length (s), $g_{b}$ is effective green time for the bicycle lane (s), $V_{eb}$ is the e-bike flow rate (bike/h), $V_{b}$ is the bicycle flow rate (bicycle/h), $C_{b}$ is capacity of the bicycle lane (bicycle/h), and $d_{eb}$, $d_{b}$ are the delay of e-bike and bicycle.

Then, the delay model integrated e-bikes with bicycles
(5)$$d=0.798\times d_{eb}+0.309\times d_{b}$$

## 4. Methodology for LOS Model of the Non-Motorized Vehicle at Signalized Intersection

Crossing facilities, traffic conflicts, and delay act as the three main factors which pose an impact on the LOS of the non-motorized vehicles at the signalized intersection. The proposed delay model of the non-motorized vehicle could be used to calculate the average delay at signalized intersections. After choosing the significant factors influencing LOS, statistical methods were used to select the interdependence between model variables and explanatory variables to establish the integration of multiple variables, and thus remove those insignificant variables.

### 4.1. Questionnaire of Rider’s Perceptions

Traffic conditions, crossing facilities, and delays were contained in the site investigation at the signalized intersection. In the field investigation, investigators rode e-bikes and bicycles to cross the intersections to obtain actual perception data. The experimental data were used to verify the respondents’ performance.

The video was reviewed and 20 short videos were selected according to the service levels. Then, the volunteers reviewed the videos and assigned a score according to the service levels A–F. Before they watched the video and made evaluations, the researchers explained to the participants the meaning of LOS, and then asked them to ride an electric bicycle or bicycle through the intersection themselves to feel the level of comfort and safety of crossing the intersection. Then, they were asked to use the pre-quantified evaluation parameter standard to express their thoughts. The six letters A–F are used to correspond to six different grades, which are used to express six different street crossing feelings of “excellent”, “good”, “average”, “inferior”, “ poor”, and “very poor” (for example: LOS A = 1, excellent). Considering the operation attributes and their cycling feelings, the intersection is evaluated from the following factors: (1) ride space for the non-motorized vehicles (left and right and front); (2) ability to pass or overtake during cycling; and (3) impact of right-turning motor vehicles. The questionnaire was completed according to the participants’ personal feelings of subjective evaluation, using the six grades of A to F, respectively, on behalf of their riding experience perception. The questionnaire is shown in Appendix A, Table A1.

A total of 400 respondents participated in the survey, 178 females and 222 males. Table 3 shows the gender and age distributions of each survey site and the average user score.

### 4.2. Representation of Survey Results by a Single LOS Grade

When selecting single-value to analysis research data, the average and mode are usually selected. For this study, selecting the mode can intuitively reflect the service level of the intersection from the perspective of participants, but the mode may not be unique, that is, one intersection corresponds to two LOS, which is very embarrassing for the research results, so it is not mentioned in the article. Although the average does not have two values at the same time, it is vulnerable to extreme values, and for this study, there are almost no intersections with LOS A or LOS F. Take LOS A as an example, to be identified as LOS A required every participant to give the same evaluation of the intersection. Even if a large number of participations chose LOS A, it takes only a few different responses to switch the final result to another one (Table 4).

The rows labeled with LOS 1, LOS 2, and LOS 3 were filled with letters instead of percentage values like the first six lines because of the new fitting approaches. The three new LOS correspond to three different parameters: straight thresholds, thresholds shifted to midpoints, and compressed ranges (Table 5). Using the average value of the six different distributions calculated before, we listed the service levels corresponding to the range of the average value one by one. Then, for each newly defined LOS, LOS A to LOS F can be represented for each corresponding distribution. This means that the six letters from A to F and the six distributions from 1 to 6 are in one-to-one correspondence.

Observing the LOS 1 line, it can be seen that the LOS obtained by replacing the corresponding range of the straight threshold has reached the expected D, E, and F levels for the three distributions of 4, 5, and 6, which also shows that the straight threshold method is not suitable for the three distributions of 1, 2, and 3.

In the same way, for the row labeled LOS 2, which adopts the thresholds shifted to midpoints, the distributions of 4 and 5 have reached the expected level; the distributions of 2 and 3 were improved under this parameter and reached the expected level, but unfortunately, distribution 1 and 6 still performed poorly.

The LOS 3 line that used compressed ranges compresses the range of LOS B to LOS E, leaving more room for LOS A and LOS F. In other words, the range of these two extreme LOS has a higher tolerance which can ensure that a large number of values at the two extremes can be covered by them correctly.

The field data of the non-motorized vehicle was used to verify the LOS 3 threshold approach which led to the conclusion that this approach could produce a reasonable range of LOS. Therefore, the data collection results were reported by this threshold approach.

### 4.3. Linear Regression Tests

To evaluate the average score obtained by each respondent in the field survey, it is necessary to explore the linear regression technique to determine whether there may be multiple linear relationships.

For the selection of variables in linear regression problems, independent variables are used to evaluate the dependent variables preliminarily selected from a large number of explanatory variables, and the variables irrelevant to the dependent variables are eliminated. The positive stepwise regression method is used to cover the important variables that can improve the prediction ability of the dependent variables of the model and *R^2^* value is used as the proxy of that capability. The adjusted *R^2^* value for the overall model is 0.367, which means only 36.7% of the variation in mean participant ratings could be estimated by the model. In addition, the coefficients were found to be significant at the 95% confidence interval. The results of the stepwise multiple linear regressions are shown in Table 6.

The best-fit model is the mean LOS score of the non-motorized vehicle which can be calculated by:(6)$$a_{1}\times \ln\left(Q_{eb}\right)+a_{2}\times \ln\left(Q_{b}\right)+a_{3}\times V_{b}+a_{4}\times \ln\left(C_{rv}\right)+a_{5}\times \ln\left(C_{p}\right)+a_{6}\times d+a_{7}$$
where

$Q_{eb}$,$Q_{b}$ are the volume of electric bicycles and bicycles, respectively,$V_{b}$ is the average speed of bicycles passing through,$C_{rv}$ is the number of conflicts between non-motor vehicles and pedestrians,$C_{p}$ is the number of conflicts between non-motor vehicles and right-turning motor vehicles, andd is the delay of non-motorized vehicles passing the intersection calculated by the delay model.

It can be seen from the above model that when the number of non-motor vehicles increases, the LOS decreases. However, at most intersections in China, bicycles account for a small proportion of non-motorized vehicles while electric non-motorized vehicles account for a large portion, so the number of bicycles has little impact on the LOS. Because the e-bikes usually tend to cross the intersections in a group, there is certain collinearity between the e-bikes’ flow and speed. The speed of e-bikes is eliminated in the model. Besides, because the right-turning vehicles are not controlled by signal lights, and tend to slow down to wait for the non-motorized vehicles to advance, therefore, the impact of motor vehicles on non-motor vehicles passing through the intersection is small. The number of conflicts between non-motorized vehicles and pedestrians and the delay of non-motorized vehicles passing through intersections is closely related to the LOS score.

### 4.4. Limitations of Linear Regression Model

When modeling ordered variables, the linear regression model may not be the best choice. It cannot deal with the situation of more than one dependent variable and the multicollinearity between independent variables, nor can it measure some variables that cannot be measured directly. As for this study, they are variables with strong subjectivity.

In order to avoid these limitations, we also use the cumulative logistic regression method to predict the response probability corresponding to each LOS according to the combination of explanatory variables, and can make full use of the 400 data samples obtained from the field survey, rather than just using the average value of 20 entrances to evaluate the LOS of the intersection.

### 4.5. Cumulative Logistic Regression

Since the quantified LOS index is discrete rather than displaying continuous variables, the linear regression model cannot analyze the data well, so we turn to cumulative logistic regression to solve the problem.

The cumulative probability P(Y≤j|x) is defined as follows:(7)$$\ln\frac{P\left(Y\leq j|x\right)}{1-P\left(Y\leq j|x\right)}=\alpha^{\prime }-\beta^{\prime }\left(x\right)$$
where

P is the probability,γ is ordered variables,j is the score of LOS, andx are explanatory variables of the influential factors.

Under the general circumstances, P(Y≤j|x)=1−P(Y≤j−1|x), so it is not hard to obtain estimated probabilities for all scores. Vector $\beta^{\prime }$ denotes the vector of coefficients for both LOS ranges and the coefficients of the independent variables considered in the model. Therefore, we modified Equation (8) as follows:(8)$$P\left(Y\leq j|x\right)=\frac{\exp\left(\alpha^{\prime }-\beta^{\prime }\left(x\right)\right)}{1+\exp\left(\alpha^{\prime }-\beta^{\prime }\left(x\right)\right)}$$

Each cumulative probability has its own intercept $\alpha_{j}$, with the fixed x, $\alpha_{j}$, and P(Y≤j|x) showing the same upward tendency. For each j within the model, it is assumed that all effects caused by $\beta_{Q_{b}}$, $\beta_{V_{eb}}$, $\beta_{V_{b}}$, $\beta_{Q_{v}}$, $\beta_{C_{rv}}$, $\beta_{C_{p}}$, and $\beta_{d}$ are equal. In order to make an understandable explanation, we consider the value for $Q_{b}$, $V_{eb}$,$V_{b}$, $Q_{v}$, $C_{rv}$, $C_{p}$, and d as 0 is expected for two scores, j and k, with j < k. After some algebraic operations, the result is as follows:(9)$$P\left(Y\leq k|Q_{eb}\right)=P\left(Y\leq j|Q_{eb}+\left(\alpha_{k}-\alpha_{j}\right)/\beta \right)$$

As shown in Figure 5, the value of each $\alpha_{j}$ increases and the value of β is positive for the numbers of the e-bikes. Table 6 illustrates the model coefficients and their significance.

The increasing value of the intercept ensures that each value of rating-num is an integer and the order of cumulative probabilities for a certain value, l, of the number of non-motorized vehicles is in the right order, meaning that:(10)$$\begin{array}{l} P\left(Y\leq 1|Q_{eb}=l\right)\\ \leq P\left(Y\leq 2|Q_{eb}=l\right)\\ \leq P\left(Y\leq 3|Q_{eb}=l\right)\\ \leq P\left(Y\leq 4|Q_{eb}=l\right)\\ \leq P\left(Y\leq 5|Q_{eb}=l\right)\leq 1 \end{array}$$

As shown in Figure 5, when the number of e-bikes ($Q_{eb}$) increases, riders are more likely to choose a low LOS. For instance, the value P(Y=1)=P(Y≤1) is higher when ln (number of the e-bikes) = 0.5 than when ln = 1, and the value P(Y=5)=P(Y≤5)−P(Y≤4) is higher when ln = 1 than ln = 0.5. Therefore, it seems that crossing with e-bikes is more likely to get a higher LOS score.

## 5. Recommended LOS Model of the Non-Motorized Vehicle

The LOS model of non-motorized vehicles provided above evaluates the average satisfaction level of non-motorized vehicle drives with the intersection facilities, in which LOS A means “excellent” and LOS F represents “terrible”.
LOS of the non-motorized vehicle = mean (LOS)(11)

The mean LOS rating can be calculated by Formula 13 below,
(12)$$\mathrm{mean}\left(\mathrm{LOS}\right)=\sum_{J=1}^{6}P\left(\mathrm{LOS}=J\right)\times J$$

The mean LOS number is converted to the average letter grade for the facility (LOS 3 in Table 4 and Table 5) and there are differences between the numerical threshold used for converting the average score to the average LOS letter grade and the score (J) used to calculate the average score.

The probability that a participant will accurately evaluate a given facility as LOS J is equal to the probability that removing evaluations that are worse than LOS J from the cumulative probability of both LOS J evaluations and even worse ones.
(13)P(LOS=J)=P(LOS≤J)−P(LOS≤J−1)

The cumulative logit model below gives the probability when a participant decides to rate a given facility as LOS J or worse:(14)$$P\left(Y\leq J\right)=\frac{1}{1+\exp\left(-\alpha_{\left(J\right)}+\sum_{K}\beta_{K}x_{K}\right)}$$
where

P(Y≤J) is the probability the intersection received an LOS grade J or worse,$\alpha_{\left(J\right)}$ denotes the maximum numerical threshold for LOS grade J (Table 7),$\beta_{K}$ is the calibration parameters for attributes (Table 7), and$x_{K}$ is the attributes (k) of the segment or facility (Table 7).

In order to keep the accuracy of the experimental data, the maximum likelihood estimation method is used to calibrate the threshold values ($\alpha_{\left(J\right)}$) and the attribute equation coefficients ($\beta_{K}$) by taking the paired facility features and LOS perception data into account.

## 6. Application and Performance of the LOS Model of the Non-Motorized Vehicle

The developed model is tested at the intersection with the location number 18 in Table 1. Row 18 of Table 8 shows the observed vehicle volume, the non-motorized vehicle volume, collision volume, and delay of the non-motorized vehicle.
(15)$$\begin{array}{l} mean\left(LOS\right)=\sum_{J=1}^{6}P\left(LOS=J\right)*J\\ =0.01*1+0.05*2+0.16*3+0.30*4+0.37*5+0.11*6\\ =4.3=LOSE \end{array}$$

As shown in Table 8, the linear regression LOS model matched 55% of the survey data while the logistic LOS model matched 10% higher, at 65%. All factors have a direct impact on LOS. Based on the application and comparison, the recommended logistic model performed with more accuracy.

## 7. Conclusions

Due to the complex and multidimensional environment, non-motorized vehicles are affected by many factors, which greatly affect their perception of safety, comfort, and convenience. Investigation and research on these factors are necessary to evaluate the facilities of non-motorized vehicles. In addition, a standardized evaluation method is needed to understand the degree of adaptation to non-motorized vehicles at intersections.

This study helps to evaluate the factors that affect the service level of non-motorized vehicles at intersections and proposes a method to determine the LOS of non-motorized vehicles at signalized intersections under mixed traffic conditions. The LOS model recommended can accurately reflect the rider’s perception when crossing a signalized intersection by combining the safety, comfort, and operability perceived by the rider when crossing the street. The field data collected in this study included the participants’ perceptions of safety, comfort, and operability when passing through selected signalized intersections and the resulting model can measure the rider’s views on how the geometric and operational characteristics of the intersection meet their needs.

A delay model of the non-motorized vehicle suitable for mixed traffic conditions was established and verified by measured data. The signal compliance rate and the irregular arrival rate of non-motorized vehicles were considered to improve the non-motorized vehicle delay model in The 2016 Highway Capacity Manual. The average delay was found to be 26.39 s and the average compliance rate of the non-motorized vehicle was 89.68%. The error rate between the delay value predicted by the delay model proposed in this study and the actual value is only 7%, which is 35.82% lower than that predicted by the HCM. Cumulative logistic regression and linear regression methods were developed to identify the significant factors influencing the LOS of the non-motorized vehicle at signalized intersections, including vehicle traffic, non-motorized traffic, conflicts, and the delay of the non-motorized traffic. Due to the limitations of linear regression techniques, cumulative logistic regression was carried out to develop a model suitable for mixed traffic conditions in China. This model can predict the probability of responses within each LOS based on the combination of explanatory variables, and improve the quality of service while separating conflicting vehicles to accommodate cyclists comfortably and safely. Moreover, the LOS model recommended in this paper can be used in other areas of developing countries with large non-motorized vehicle flows. However, a modification is required to *K_c_* because of the differences in the proportion of violations. The LOS model of the non-motorized vehicle provides a measure of the safety and comfort of an intersection relative to the cyclist. Roadway designers can determine how well a particular intersection accommodates non-motorized travel by using the value of the LOS of the non-motorized vehicle at an intersection. This measure will help assess and premeditate the non-motorized vehicles’ travel needs of existing intersections.

The current study is not without limitations. First, traffic data during weekends and holidays were out of our consideration. However, these neglected periods were the time when riders truly prefer intersections with high LOS [19,20]. Next, although the error rate predicted by the modified model was significantly improved, there is still room for improvement. Future research can take more relevant parameters into account, such as the integrity of crossing facilities. Finally, with the development of autonomous driving technology, autonomous vehicles will not only reconstruct people’s travel mode but also affect the intersection service level [21]. Studying the conflicts or interactions between autonomous vehicles and non-motorized vehicles at intersections should be put on the agenda.

## Figures and Tables

**Figure 1 ijerph-19-04534-f001:**
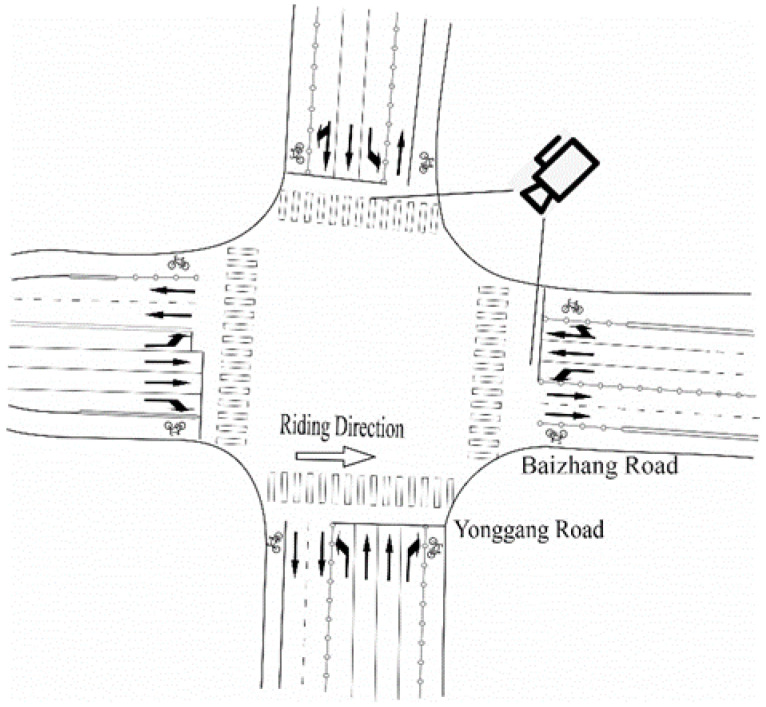
Setting details for observation.

**Figure 2 ijerph-19-04534-f002:**
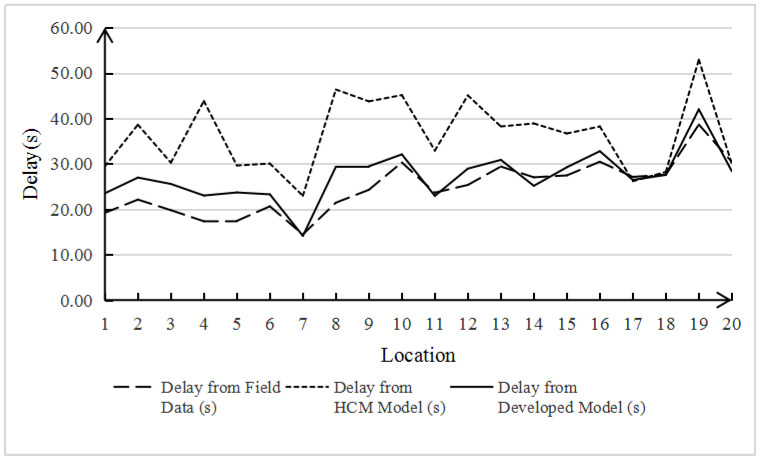
Comparative analysis of e-bike delay.

**Figure 3 ijerph-19-04534-f003:**
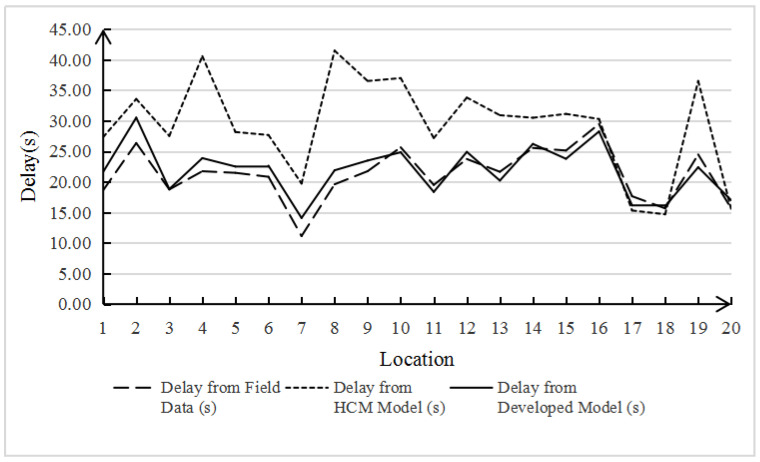
Comparative analysis of bicycle delay.

**Figure 4 ijerph-19-04534-f004:**
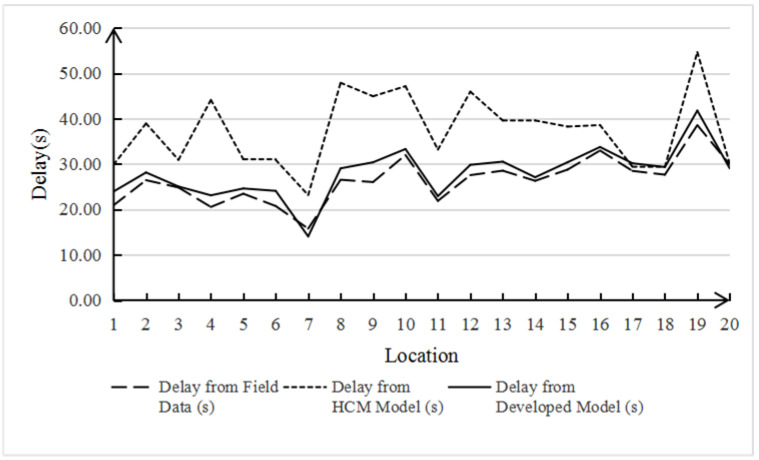
Comparative analysis of the non-motorized vehicle delay.

**Figure 5 ijerph-19-04534-f005:**
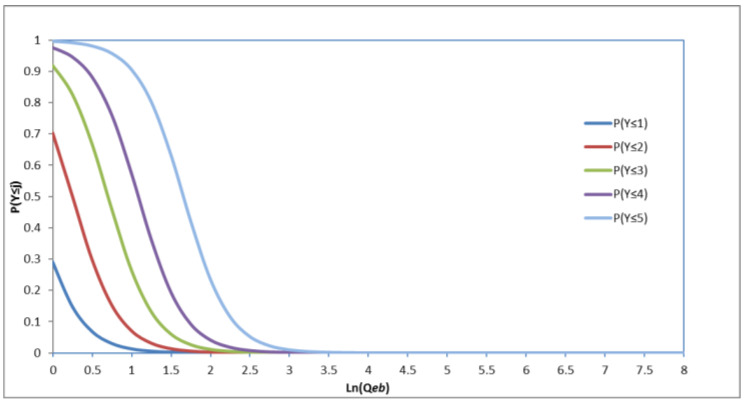
Example cumulative logit distribution of LOS.

**Table 1 ijerph-19-04534-t001:** Characteristics of Data Collection.

Location	Cycle Time (s)	Green Time (s)	Number of the Non-Motorized Vehicles Arriving during Non-Green Phase		Number of the Non-Motorized Vehicles Arriving During Green Phase		Signal Compliance Rate	
			E-Bike	Bicycle	E-Bike	Bicycle	E-Bike	Bicycle
1	150	60	179	25	157	23	0.8989	0.9122
2	150	50	257	16	223	8	0.8696	0.9090
3	150	60	240	35	192	40	0.9125	0.8795
4	150	40	146	12	190	12	0.8864	0.8636
5	150	60	161	78	151	66	0.9297	0.8854
6	150	60	214	54	194	42	0.8860	0.8733
7	125	55	271	15	389	15	0.8392	0.8000
8	150	40	219	40	237	56	0.9666	0.9290
9	185	70	405	43	475	57	0.9090	0.9302
10	185	70	498	75	502	85	0.8872	0.8909
11	150	60	335	11	385	13	0.8987	0.8847
12	150	50	417	25	495	23	0.9363	0.9433
13	150	55	538	58	422	62	0.9125	0.8571
14	150	55	488	35	532	25	0.8563	0.9333
15	150	55	472	83	338	67	0.8667	0.8744
16	150	55	580	19	380	11	0.8989	0.9326
17	125	65	738	100	612	80	0.8875	0.9112
18	125	65	775	34	695	26	0.8889	0.9231
19	185	70	842	44	718	56	0.9136	0.8667
20	125	65	978	120	972	90	0.9053	0.9217

**Table 2 ijerph-19-04534-t002:** Delay Model Validation.

Location	Delay from Field Data (s)		Delay from HCM Model (s)		Delay from Developed Model (s)	
	E-Bike	Bicycle	E-Bike	Bicycle	E-Bike	Bicycle
1	19.22	18.63	29.4760	27.3279	23.5257	21.6394
2	22.13	26.35	38.6313	33.5635	26.9800	30.5092
3	19.8	18.76	30.2691	27.5159	25.5746	18.8224
4	17.35	21.74	43.8406	40.5651	23.0260	23.8855
5	17.44	21.48	29.6424	28.1585	23.7015	22.5077
6	20.65	20.83	30.0668	27.6639	23.2875	22.6490
7	14.46	11.13	22.9688	19.7315	14.1332	14.0940
8	21.45	19.6	46.3754	41.4708	29.3570	21.8900
9	24.29	21.75	43.7672	36.5037	29.4550	23.4885
10	30.31	25.65	45.1494	36.9758	32.0905	24.8406
11	23.66	19.5	32.9268	27.1630	22.9470	18.3571
12	25.36	23.74	45.0799	33.7968	28.9488	24.9067
13	29.41	21.65	38.2415	30.9075	30.8778	20.2167
14	27.04	25.55	38.9009	30.4899	25.1636	26.2096
15	27.46	25.13	36.6870	31.1207	29.2554	23.7746
16	30.47	29.47	38.2415	30.2852	32.7922	28.2440
17	27.13	17.65	26.1818	15.3191	26.4636	16.1560
18	27.64	15.67	28.2353	14.6939	27.5669	16.0130
19	38.66	24.48	52.9530	36.5037	42.0056	22.3941
20	30.45	15.65	30.0000	15.4839	28.3777	16.9899

**Table 3 ijerph-19-04534-t003:** Gender and Age Distribution and Average User Score of Each Survey Site.

Site ID	Number of Participants	Number of Males	Number of Females	Percentage by Age Group (Years)				Average User Score
				≤20	21–39	40–59	≥60	
1	20	8	12	4	8	6	2	1.30
2	20	13	7	2	9	5	4	1.45
3	20	10	10	4	7	7	2	2.45
4	20	9	11	5	10	4	1	2.00
5	20	11	9	3	5	7	5	2.95
6	20	12	8	5	8	4	3	2.60
7	20	9	11	3	9	6	2	3.80
8	20	8	12	4	7	7	2	3.85
9	20	10	10	5	6	5	4	3.95
10	20	10	10	2	11	6	1	3.80
11	20	12	8	3	7	9	1	3.55
12	20	14	6	4	6	7	3	3.40
13	20	9	11	1	8	6	5	3.80
14	20	7	13	3	10	5	2	4.50
15	20	12	8	4	5	7	4	4.25
16	20	11	9	2	10	6	2	4.20
17	20	8	12	3	7	5	5	4.50
18	20	10	10	4	7	6	3	4.30
19	20	14	6	3	8	6	3	4.30
20	20	10	10	4	6	7	3	4.30

**Table 4 ijerph-19-04534-t004:** Representation of Survey Results by a Single LOS Grade: Distributions and Mean of LOS.

LOS	Results by Distribution					
	1	2	3	4	5	6
**A**	60%	10%	NA	NA	NA	NA
**B**	26%	67%	20%	18%	NA	NA
**C**	14%	23%	35%	25%	14%	8%
**D**	NA	10%	30%	34%	28%	19%
**E**	NA	NA	15%	23%	40%	36%
**F**	NA	NA	NA	NA	18%	37%
**1**	B	C	D	D	E	F
**2**	B	B	C	D	E	E
**3**	A	B	C	D	E	F

Note: NA = not available; mean for distributions 1–6, respectively: 1.54, 2.13, 3.40, 3.62, 4.62, and 5.02; mode for distributions 1–6, respectively: 1, 2, 3, 4, 5, and 6.

**Table 5 ijerph-19-04534-t005:** Representation of Survey Results by a Single LOS Grade: LOS Mean Value Threshold Schemes.

LOS	Numerical Value	LOS 1, Straight Thresholds	LOS 2, Thresholds Shifted to Midpoints	LOS 3, Compressed Ranges
A	1	Mean ≤ 1.00	Mean ≤ 1.50	Mean ≤ 2.00
B	2	1.00 < Mean ≤ 2.00	1.50 < Mean ≤ 2.50	2.00 < Mean ≤ 2.75
C	3	2.00 < Mean ≤ 3.00	2.50 < Mean ≤ 3.50	2.75 < Mean ≤ 3.50
D	4	3.00 < Mean ≤ 4.00	3.50 < Mean ≤ 4.50	3.50 < Mean ≤ 4.25
E	5	4.00 < Mean ≤ 5.00	4.50 < Mean ≤ 5.50	4.25 < Mean ≤ 5.00
F	6	5.00 < Mean	5.50 < Mean	5.00 < Mean

**Table 6 ijerph-19-04534-t006:** Multiple Linear Regression Model.

Model	Model Estimate	Coefficient	SE	Sig
$Q_{eb}$	$a_{1}$	2.132	0.373	0.000
$Q_{b}$	$a_{2}$	0.120	0.127	0.044
$V_{b}$	$a_{3}$	0.071	0.109	0.013
$C_{rv}$	$a_{4}$	−1.171	0.415	0.005
$C_{p}$	$a_{5}$	0.761	0.284	0.008
d	$a_{6}$	0.039	0.017	0.023
Constant	$a_{7}$	−9.906	1.139	0.000

**Table 7 ijerph-19-04534-t007:** Maximum Likelihood Estimates for LOS Model.

Parameter	Estimate	SE	Wald	DF	Sig	95% Confidence Interval	
						Lower Bound	Upper Bound
Intercept 1	19.434	2.101	85.594	1	0.000	15.317	23.551
Intercept 2	21.193	2.133	98.767	1	0.000	17.041	25.373
Intercept 3	22.743	2.176	109.193	1	0.000	18.477	27.009
Intercept 4	24.068	2.207	118.951	1	0.000	19.743	28.393
Intercept 5	26.038	2.241	134.957	1	0.000	21.645	30.431
$Q_{eb}$ $,\beta_{Q_{eb}}$	3.444	0.639	29.063	1	0.000	2.192	4.697
$Q_{b}$ $,\beta_{Q_{b}}$	0.666	0.175	14.523	1	0.000	0.324	1.009
$V_{eb},\beta_{V_{eb}}$	0.319	0.195	2.673	1	0.102	−0.063	0.702
$V_{b}$ $,\beta_{V_{b}}$	0.040	0.182	0.049	1	0.826	−0.316	0.396
$Q_{v}$ $,\beta_{Q_{v}}$	0.173	0.210	0.683	1	0.408	−0.238	0.584
$C_{rv}$ $,\beta_{C_{rv}}$	−1.796	0.692	6.734	1	0.009	−3.153	−0.440
$C_{p}$ $,\beta_{C_{p}}$	1.257	0.478	6.913	1	0.009	0.320	2.193
d $,\beta_{d}$	−0.067	0.028	5.6453	1	0.018	−0.122	−0.012

**Table 8 ijerph-19-04534-t008:** Evaluation of Proposed LOS Model of The Non-motorized Vehicle.

Survey Number	Values by Variable								Survey LOS	Linear Model LOS	Logistic Model LOS
	$Q_{v}$ **(pcu/h)**	$Q_{eb}$ **(e-bike/h)**	$Q_{b}$ **(bicycle/h)**	$C_{rv}$	$C_{p}$	$V_{eb}$ **(m/s)**	$V_{b}$ **(m/s)**	d **(s)**			
1	120	336	48	113	75	3.96	3.22	20.92	A	A	B
2	144	480	24	175	121	5.16	2.73	26.44	A	B	B
3	192	432	75	196	106	4.02	3.22	24.83	B	B	B
4	168	336	24	120	55	5.79	2.35	20.54	A	A	A
5	216	312	144	103	89	4.84	2.94	23.46	C	B	B
6	144	408	96	250	206	3.89	2.27	20.74	B	B	B
7	450	660	30	275	160	5.19	3.29	15.77	D	B	D
8	264	456	96	152	142	4.91	2.56	26.53	D	C	C
9	80	880	100	275	125	4.35	3.34	26.02	D	C	D
10	340	1000	160	390	220	5.87	4.50	31.98	D	D	E
11	168	720	24	214	150	4.91	2.50	21.88	D	C	D
12	72	912	48	394	122	4.85	2.80	27.55	C	C	C
13	390	960	120	374	176	4.94	4.03	28.53	D	D	D
14	120	1020	60	360	203	4.14	3.63	26.30	E	D	D
15	450	810	150	290	130	5.01	3.64	28.77	D	C	D
16	240	960	30	268	182	4.15	2.90	32.97	D	D	C
17	180	1350	180	450	202	4.04	3.25	28.48	E	E	E
18	360	1470	60	481	214	3.93	2.22	27.65	E	D	E
19	400	1560	100	441	179	4.98	2.65	38.57	E	E	E
20	270	1950	210	584	256	3.61	2.46	29.88	E	E	E

## Data Availability

All the data used in this research were collected by questionnaires and presented in Table 1.

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
