# Peer review of "Level of Service Model of the Non-Motorized Vehicle Crossing the Signalized Intersection Based on Riders’ Perception Data"

_ijerph, 2022, doi:10.3390/ijerph19084534_

Round 1

Reviewer 1 Report

Indeed, increasing the number of vehicles leads to increased congestion, delays, reduced speed, so a low level of service, LOS. The authors used as a method of data collection, video recording during peak hours, morning and evening. The paper does not show how many times the data were collected from the intersections. What is the value of green time for non-motorized vehicles?

The authors add in the model of delays in the HCM manual, two factors, namely:

  • rule compliance degree KC
  • irregular arrival rate KNU

How did the new model come about? Just by adding?

But until the end, the results of the research are well-defined and highlighted.

The paper provides a mathematical model that can use the calculation possibilities and tools necessary to simulate quite complex situations, close to real situations.

In conclusion the paper is  good in terms of scientific contribution.

Reviewer 2 Report

Overall, the research is well-designed and the draft is well-structured. There are some minor revisions I would propose:

The abstract should be more clear and more precise. In my opinion, it does not summarize the paper properly. For now, it's very confusing and does not show a clear picture of what was done in the paper.

E.g. this sentence is too long and confusing. Should be split in two. "Aiming at the mixed non-motorized traffic flow of electric vehicles and bicycles in Chinese 16 metropolis, through questionnaire design and field survey data, the delay model in The Highway 17 Capacity Manual is modified by taking two new factors which are the pedestrian traffic rule compliance rate and the fuzzy perception of arrival rate in reality into account. "

In general, the introduction is well-written. It provides a decent insight into the main subject of the research. However, the concept of LOS should be defined before the State of the art section. I advise the authors to do it in the introduction section because they are mentioning LOS more than 10 times in the introduction before it was described in detail in the Literature review section.

Table 1 should not be placed on two separate pages. Please keep it on one page.

Could you tell us more about the method the cameras use for the detection of conflicts related to non-motorized vehicles crossing the street? This is very important input data, thus should be well described and presented.

The conclusion is very short compared to the number of presented results. You should highlight more results in the conclusion. There are lots of interesting results to highlight and underline in the conclusion.

Reviewer 3 Report

Dear authors/editor,

The overall quality of the work is good. The topic is interesting and actual since the non-motorized and e-motorized two-wheel traffic demand tends to increase in the cities worldwide. The research conducted and the model proposed for measuring control delay and LOS in urban intersections are restricted to China cities only, which may be considered as the model limitation. Rider’s perception of traffic and traffic environment in other geographic areas may be different; therefore, the performance of the proposed LOS model values may vary for similar traffic conditions.

The question to the authors is: To what extent the calibration parameters can be adjusted to adapt the model for different traffic behaviors? Please include this consideration in the manuscript.

Here are some other remarks and questions to authors:

  • In the HCM model for vehicle delay, the effective green (g) includes the intergreen intervals, while the green phase suggests only a green interval is included. That means that vehicles can pass the intersection on a “non-green” signal in some circumstances. Therefore, the impact on the control delay for non-green passing is already included in HCM. Hence the purpose of the KC parameter (compliance with traffic rules) needs better explanation and justification.
  • On which basis the Capacity of the bicycle lane has been calculated? What is the Saturation flow rate used for calculation for selected intersections? Please explain in the manuscript (chapter 3.2).
  • Control delay in peak-period may significantly impact traffic conditions. For PHF factor determination, 15-min periods are representative. It is not evident that you used PHF for the Control Delay calculation. Please explain and refer to the manuscript on Peak-factor impact for non-motorized vehicles.
  • When comparing the Control Delay for different traffic situations, it is helpful to know the degree of saturation for the corresponding movements of the intersection (in this case, for the b-lane/movements).

Please consider these questions/comments and update the manuscript accordingly. Also, in the Conclusion section, list not only the valuable contribution of the research but also look at some weak points of the model and indicate its limitation.
